# Enhanced Removal of Endocrine-Disrupting Compounds from Wastewater Using Reverse Osmosis Membrane with Titania Nanotube-Constructed Nanochannels

**DOI:** 10.3390/membranes12100958

**Published:** 2022-09-30

**Authors:** Nor Akalili Ahmad, Pei Sean Goh, Nurfirzanah Azman, Ahmad Fauzi Ismail, Hasrinah Hasbullah, Norbaya Hashim, Nirmala Devi Kerisnan@Krishnan, Nasehir Khan E. M. Yahaya, Alias Mohamed, Muhammad Azroie Mohamed Yusoff, Jamilah Karim, Nor Salmi Abdullah

**Affiliations:** 1Advanced Membrane Technology Research Centre (AMTEC), School of Chemical and Energy Engineering, Faculty of Engineering, Universiti Teknologi Malaysia, Skudai 81310, Malaysia; 2National Water Research Institute of Malaysia (NAHRIM), Lot 5377, Jalan Putra Permai, Seri Kembangan 43300, Malaysia; 3Sewerage Service Department (JPP), Block B, Level 2 & 3, Atmosphere PjH No 2, Jalan Tun Abdul Razak, Precinct 2, Putrajaya 62100, Malaysia

**Keywords:** endocrine-disrupting compound, thin-film composite membrane, titania nanotube, bisphenol A, caffeine

## Abstract

This paper presents a comprehensive study of the performance of a newly developed titania nanotube incorporated RO membrane for endocrine-disrupting compound (EDC) removal at a low concentration. EDCs are known as an emerging contaminant, and if these pollutants are not properly removed, they can enter the water cycle and reach the water supply for residential use, causing harm to human health. Reverse osmosis (RO) has been known as a promising technology to remove EDCs. However, there is a lack of consensus on their performance, especially on the feed concentrations of EDC that vary from one source to another. In this study, polyamide thin-film composite (PA TFC) membrane was incorporated with one-dimensional titania nanotube (TNT) to mitigate trade-off between water permeability and solute rejection of EDC. The characterization indicated that the membrane surface hydrophilicity has been greatly increased with the presence of TNT. Using bisphenol A (BPA) and caffeine as model EDC, the removal efficiencies of the pristine TFC and thin-film nanocomposite (TFN) membranes were evaluated. Compared to TFC membrane, the membrane modified with 0.01% of TNT exhibited improved permeability of 50% and 49% for BPA and caffeine, respectively. A satisfactory BPA rejection of 89.05% and a caffeine rejection of 97.89% were achieved by the TNT incorporated TFN membranes. Furthermore, the greater hydrophilicity and smoother surface of 0.01 TFN membrane led to lower membrane fouling tendency under long-term filtration.

## 1. Introduction

Micropollutants and anthropogenic contaminants such as endocrine-disrupting compounds (EDCs) and pharmaceutically active compounds have been increasingly detected in various water sources [1]. The abundance of these pollutants is closely associated with their inevitable usage in a wide range of industries. For example, bisphenol A (BPA) is extensively used in many consumer products, such as epoxy resins and polycarbonated plastics [2]. Undesirably, upon the exposure to elevated temperature, BPA can leach from these polymers into the food and water [3]. Besides, it can be produced by the acid catalyzed condensation of phenol and acetone under a mild condition of temperature and pressure [4]. Similarly, caffeine, which has been commonly used in pharmaceutical and food products, has also been traced in wastewater and surface water [5]. Like most pharmaceutically active compounds, caffeine residues exert negative impacts on ecological and human safety. It causes mutagenesis that leads to deoxyribonucleic acid (DNA) damage and toxicity to aquatic organisms [6]. Due to the enormous negative impacts imposed by EDC pollutants on the environment and human health, researchers have put strenuous efforts into developing a sustainable solution to remove these micropollutants from various water bodies. Unfortunately, compared to most organic substances, micropollutants are normally found in relatively low amounts in aqueous environments [7], hence increasing the difficulty to control and eliminate these compounds during the typical wastewater treatment processes [8]. Some micropollutants can be partially removed via conventional treatment methods through vaporization or sorption onto activated sludge, such as fragrances, but the production and disposal of sludge in high volumes remain cumbersome [9].

Many recent membrane research investigations have focused on applying membrane-based separation processes, particularly reverse osmosis (RO) and nanofiltration to remove EDC. Most contaminants present at low concentrations or dissolved in colloidal form can be effectively removed with high selective control at a moderate temperature [5]. Thin-film composite (TFC) RO membrane has also been widely used in wastewater treatment owing to their high water permeability, durability and great selectivity [10]. Khazaali et al. utilized commercial RO membrane for BPA and achieved a removal efficiency of 87% at a feed concentration of 50 mg/L [4]. Boleda et al. achieved a caffeine (1 mg/mL concentration) removal efficiency between 74 and 83% by using commercial RO membrane [11]. The effective EDC removal through membrane separation processes has been a challenge mainly due to the low concentration of EDC in the water body (<10 µg/L) [12]. The intrinsic constraints of membrane-based processes have further limited their applications. For decades, constant efforts have been made to address the trade-off between water permeability and solute rejection. Furthermore, fouling issues in membrane processes must also be suppressed for practical long-term application [13].

Numerous materials are developed for wastewater treatment [14,15]. The use of nanomaterials in the development TFC membrane has opened the opportunities to improve the performance of the RO membrane in terms of water permeability, antimicrobial properties, fouling and chlorine resistance of the membranes, as well as thermal stability and mechanical strength [16]. Various one-dimensional nanostructures such as titania nanotube (TNT) and carbon nanotube (CNT) have been explored for water purifications [17]. CNT has gained popularity, as it can strengthen and modify the surface chemistry of membranes [18]. Similar to titanium dioxide (TiO_2_) nanoparticles, TNT has attracted tremendous interest for water filtration application, as it is mainly based on the abundance and chemical stability [19,20,21]. The cylindrical nanostructure offers a frictionless transport of water molecules to provide the enhancement of water permeability. TNT has been increasingly explored for wastewater treatment owing to its higher hydrophilicity, larger surface area, larger pore volumes and unique cylindrical shape with open ends as compared to the conventionally used zero dimensional TiO_2_ [18,22]. TNT in the PA layer provides an additional water path for water molecules to flow through the nanochannel, hence enhancing water permeability. The oxygenated and hydroxyl functionals groups retained on the tubular structure TNT could also increase the rate of water transport. The TFC membrane with TNT incorporated as nanofiller has a great propensity to enhance the flux and antifouling properties. Khoo et al. reported the preparation of TNT incorporated nanofiltration membrane that can enhance water flux by 16% without compromising NaCl rejection [23]. The TFN membrane also showed a high flux recovery rate of 85.77%.

The present work is focused on the development of TFN RO membrane incorporated with TNT in the PA layer. The surface morphology, roughness, hydrophilicity of TFN membrane and pristine TFC membrane were characterized. The performances of membranes in terms of water permeability and EDC rejection were evaluated. BPA and caffeine were chosen as model EDCs in this study in view of their widespread application and abundance in water bodies. The long-term antifouling performance of the membranes were also assessed.

## 2. Materials and Methods

### 2.1. Materials

Polysulfone (PSf) Udels P3500 polymer in pellet form (Solvay), Polyvinylpyrrolidone (PVP, K30, Acros Organic, New Jersey, NJ, USA) and N-Methyl-2-pyrrolidone (NMP, 99.5%, Acros Organic, Billerica, MA, USA) were used for the substrate formation. TNT was formed by using commercial TiO_2_ P25 Degussa nanoparticles (Evonik Industries, Essen, Germany), hydrochloric acid (HCl, Billerica, MA, USA), sodium hydroxide (NaOH) (Merck & Co, Billerica, MA, USA) and deionized (DI) water. PA layer atop PSf substrate was formed using m-phenylenediamine (MPD, 99.0%, Merck, Billerica, MA, USA) and 1,3,5-benzenetricarbonyl trichloride (TMC, 98%, Acros Organic, Waltham, MA, USA) in n-hexane (C6H6, 49.0%, Merck, Billerica, MA, USA). NaCl (Merck) was used to evaluate the salt removal efficiencies of the membranes. BPA and caffeine used for the preparation of synthetic EDC-containing feed solution were purchased from Sigma Aldrich, Billerica, MA, USA.

### 2.2. Fabrication of TFC and TFN Membranes

The preparation of TNT was adopted from the methods reported by Subramaniam and co-workers [24]. Commercial TiO_2_ P25 Degussa nanoparticles were used as the starting material to synthesize TNT. Both HCl and NaOH were used in the synthesis of TNT. DI water was used for the washing step in TNT synthesis. In this experiment, the concentration of NaOH used was set at 10 M and temperature at 180 °C, while the hydrothermal reaction time in autoclave was 24 h. First, 3 g of TiO_2_ was added into 120 mL of 10 M NaOH and stirred for 4 h. The suspension was then transferred into a Teflon-lined autoclave container, firmly locked and placed into an oven at 180 °C to undergo hydrothermal treatment for 24 h duration. Specifically, TNT was synthesized from the alkaline hydrothermal of commercial TiO_2_ P25 precursor in a sealed heated solution under controlled conditions. The formation of TNT tubular structure from TiO_2_ nanoparticles has been thoroughly documented in current studies [24]. In brief, a strong alkali solution (NaOH) is used as a ‘bridging agent’ to attach TiO_2_ nanoparticles together. The high temperature and pressure in the autoclave condition induced the formation of nanosheets by facilitating the rolling of sheets to form a tubular structure in the presence of high saturated vapor pressure. The product formed was treated with 1 L of 0.1 M HCl overnight and then washed with DI water using a 5 L bottle till the pH of the decanter water reached 7. HCl as acid treatment was performed to remove excess Na^+^ ions, where it was substituted with H^+^ ions. The sample was then filtered, dried at 60 °C, and ground to fine particles. 

For the membrane preparation, the commercial microporous PSf support was washed with DI water for 24 h to remove residual solvent. After that, the membranes were rinsed by pure water and then soaked in pure water for at least one day before use. PA active layer was formed on top of the PSf substrate via interfacial polymerization method using 2 *w/v%* of MPD in DI water and 0.1 *w/v%* TMC in n- hexane as illustrated in Figure 1a. First, the substrate membrane was fixed between a glass plate and a rubber frame. Then, the MPD solution was poured into the setup to saturate the exposed side of PSf with the diamine monomer. After 1 min, excess MPD solution was drained off and the leftover droplets were remove using a rubber roller. Next, TMC solution was dripped onto the MPD saturated surface to initiate the interfacial polymerization process. After 50 s, excess TMC solution was poured out and the resulted TFC was left to dry in ambient temperature for 1 min before being thermally treated in oven at 60 °C for 5 min. Lastly, the membrane was stored in DI water for 24 h before further use. For the fabrication of TFN membrane, the concentration variation from 0.01, 0.05 and 0.10 *w*/*v*% of TNT were dispersed into the TMC solution as shown in Figure 1b. The mixture was then sonicated for 1 h prior to interfacial polymerization to avoid agglomeration of the nanoparticles. The final product was denoted as 0.01 TFN, 0.05 TFN and 0.1 TFN membranes. 

### 2.3. Characterizations

Transmission electron microscopy (TEM, Hitachi HT770, Redding, CA, USA operated at 120 kV) was used to examine the morphological structure of TNT. Field emission scanning electron microscopy (FESEM, Zeiss Crossbeam 340, Hitachi SU8020, Tokyo, Japan) was used to examine the surface and cross-sectional morphology of the membranes. Additionally, 0.5 cm × 0.5 cm of TFC and TFN membranes were prepared by cryo-fracturing the membrane inside liquid nitrogen to ensure a smooth cross-section break. The fractured samples were mounted on carbon tape, masked on stainless steel stub and subsequently undergo sputter coating. The FESEM examination was carried out with different magnifications of 10,000× and 50,000× to obtain different surface and cross-sectional FESEM micrographs. In this study, energy dispersive X-ray analysis (EDX, QUANTAX 70, Hitachi, Tokyo, Japan) was used to verify the distribution of Ti element in the TNT-PA TFN membrane. The membranes were scanned at magnification of ×10 k.

X-ray photoelectron spectroscopy (XPS, Thermo Fisher Scientific, Santa Barbara, CA, USA) with Kratos Axis Ultra DLD with aluminum Kα (1486 eV) radiation was used to determine the elemental composition of composite membrane. From the atomic concentration of elements at the membrane surface, the degree of cross-linking of the PA layer was obtained based on Equation (1), as reported in previous studies [25,26].

Degree of cross-linking(%) = m/(m + n) × 100%
(1)

where m is the cross-link of the PA layer and n is the linear parts of the PA layer. The values of m and n can be calculated based on the experimental O/N ratio obtained from XPS analysis using Equations (2) and (3).

m + n = 1
(2)


O/N = (3m + 4n)/(3m + 2n)
(3)


In order to observe the surface area, pore volume and pore size distribution of the TNT, Brunauer–Emmett–Teller (BET) analysis was performed with pure nitrogen gas as the absorbate. An Autosorb-1 automatic analyzer was used after degassing the samples at 453K for at least 6 h under vacuum. X-ray diffraction (XRD, Cu Kα radiation, λ = 0.154 nm, D/max-rB 12 kW Rigaku, TX, USA) to examine the crystallinity of TNT. The interlayer distance (d-spacing) of both TNT was calculated by using Bragg’s Equation (4):
d = λ_XRD_/(2 sinθ)
(4)

where λ is the wavelength of X-ray beam (0.154 nm), d is the spacing between adjacent sheets, and θ is diffraction angle. Atomic force microscopy Nanowizard 3 (AFM, JPK Instrument-contact mode, Berlin, Germany) was used to study the surface roughness (Ra) of the five membranes (10 μm × 10 μm each). The water contact angle of membrane was measured by a contact angle goniometer (OCA, 708381-T,LMS Scientific, Filderstadt, Germany). The functional groups present in TNT, pristine TFC membrane and TFN membrane were analyzed by using attenuated total reflection Fourier transform infrared spectroscopy (ATR-FTIR, Thermo Nicolet Avatar 360, Ramsey, NJ, USA).

### 2.4. Evaluation of TFC and TFN Membrane Performance

#### 2.4.1. Feed Water Characteristics

We prepared 2000 ppm of NaCl solution by dissolving 2 g of NaCl in 1 L RO water as a model of seawater to identify the effectiveness of the fabricated RO membranes for RO process. Moreover, stock solution of BPA and caffeine were utilized to represent the EDC that was commonly detected in the industrial wastewater. Table 1 summarizes the properties of the micropollutants used in this study. The BPA solution was prepared in a small amount of acetonitrile to obtain 10 ppm of BPA feed solution. Meanwhile, 0.01 g of caffeine was dissolved into 1 L RO water to prepare 10 ppm of caffeine.

#### 2.4.2. Water Permeability and Rejection Test of Saline Water

A dead-end cell (HP4750, Sterlitech Corp., Kent, UK), as illustrated in Figure 2, with an effective area of 14.6 cm^2^, was used to evaluate the RO membrane performance. Specifically, a circular membrane disc was cut and mounted in a permeation cell. A porous stainless-steel brace was used as membrane supporter and tightened by rubber O-ring. At 16 bar, the membrane was compacted to achieve a steady water permeability for 30 min. Then, at 15 bar, the membrane was stabilized with RO water for 15 min. After that, the 1 mL of RO water was collected and time taken was recorded. The filtration was repeated using 2000 ppm NaCl as feed water. The time taken to collect 1 mL of permeate was recorded. Water permeability of salt (A_w,s_) and salt rejection (R_s_) were calculated using Equations (5) and (6), respectively: 
A_w,s_ (L·m^−2^·h^−1^·bar^−1^) = V/At∆P
(5)

where V is the total amounts of the collected permeate (L), A is the cross-sectional area (m^2^) and t is duration of treatment (h).

R_s_ (%) = 100 × (1 − (C_p,s_/C_f,s_))
(6)


C_f,s_ is salt concentrations of feed and C_p,s_ salt concentration of permeate by a conductivity meter.

#### 2.4.3. Water Permeability and Rejection Test of BPA and Caffeine Solution

The BPA and caffeine solutions were prepared as described in Section 2.4.2. The collected permeate was measured by using UV-vis spectrometer (DR2800, Hach). The UV-vis absorbance was measured at 276 and 273 nm for BPA and caffeine, respectively. The EDC permeability (A_w,EDC_) and rejection (R_EDC_) were calculated using Equations (7) and (8): A_w,EDC_ (L·m^−2^·h^−1^·bar^−1^) = V/At∆P(7)
where V is the total amounts of the collected permeate (L), A is the cross-sectional area (m^2^) and t is duration of treatment (h).
R_EDC_ (%) = 100 × (1 − (C_p,EDC_/C_f,EDC_))(8)
where C_p,EDC_ and C_f,EDC_ represent concentration of EDC permeate and EDC feed (abs) respectively.

#### 2.4.4. Antifouling Assessment

The antifouling test using TFC and TFN membranes was examined using the RO test unit, as shown in Figure 2. The filtration was then carried out using 10 ppm of BPA over 540 min. The permeate was collected at interval of 30 min and the time taken was recorded. The fouling behavior of each membrane was investigated by observing the water permeability decline patterns of the pristine TFC and modified TFN membrane. The same procedure was used for the caffeine solution.

## 3. Result and Discussion

### 3.1. Properties of Titania Nanotube

The formation of TNT was confirmed by several characterizations, as shown in Figure 3. As shown in the TEM image in Figure 3a–d, the synthesized nanotubes were found in a tubular shape with open-ended lumens. The average diameter of the lumens was ~75 nm. Figure 3e presents the XRD pattern of TNT. The XRD peaks at 2θ of 10.8, 25 and 48.5° correspond to the (200), (100) and (020) planes, respectively, indexed to TNT, which is in good agreement with the previous report [24]. Moreover, these peaks correlated with the hydrogen titanate (H_2_Ti_3_O_7_) structure that represents the high amount of OH groups [27]. These three peaks indicate that the nanotubes formed are in anatase crystallinity. Furthermore, to further confirm the formation of TNT, ATR-FTIR spectra were interpreted from Figure 3f. The broad ATR-FTIR band between 3000 and 3500 cm^−1^ can be ascribed to the OH stretching, which indicates the presence of a high amount of OH groups in TNT. The vibrational peak around the 669 cm^−1^ region was attributed to the stretching vibrational mode of Ti-O-Ti. 

Particle size plays an important role in determining the final particle surface area. The surface area of TNT was analyzed as 33.58 m^2^/g, which was higher than that of commercial TiO_2_ P25 due to the tubular structure and N_2_ adsorption-desorption isotherms was shown in Appendix A. The surface area of commercial TiO_2_ P25 commonly reported in the literature is the range between 6 and 15 m^2^/g [28,29,30]. However, the surface area of the TNT synthesized in this study is smaller than that reported by previous studies using the similar hydrothermal method, which falls in the range of 50–200 m^2^/g [24,31]. The TNT synthesized in this study also possessed relatively low pore size and pore volume. This observation can be related to the presence of a high number of functional groups at the pore entrance, as based on a previous study [32]. The pore volume with small average pore size indicates more pores with smaller size were distributed along the tubular structure. The inner diameter of TNT was determined as 19.53 nm, in agreement with that of reported previously, which was around 14 and 16 nm [24,33]. 

### 3.2. Characteristics of TFC and TFN Membranes

In the subsequent section, the separation performances of TFC and TFN membranes prepared in this study are benchmarked with that of commercial TFC membranes. Therefore, in this section, the characteristics of commercial TFC membranes are selectively discussed to provide better correlations between the separation performances and their physico-chemical properties. The FESEM surface and cross-sectional morphologies of the TFC and TFN membranes are shown in Figure 4. The leaf-like structure, the typical feature of PA surface, can be clearly seen from the surface images of all TFC and TFN membranes. As shown in Figure 4a,b, the surface of commercial TFC membrane is characterized by a larger leaf-like structure, as compared to the home-fabricated counterpart. The structure of the PA layer was influenced by the hydrophilicity and pore size of the support layer [34]. Upon the embedment of TNT within the PA layer of all the TFN membrane, it was observed that the surfaces were covered with some tubular-structured materials, as shown in Figure 4c,d. This indicates the successful incorporation of TNT in the PA layer. Chong et al. also observed the deposition of functionalized TNT in the PA layer for RO membrane by using the same deposition technique [35]. 

The cross-section of TFC and TFN membranes (Figure 4A–E) exhibited a typical morphology of PA membrane. The PA layer comprised a thin interfacially polymerized PA layer at the top and a porous PSf layer at the bottom. The PA layer formation can be confirmed by the ridge and valley structure that formed over the PSf layer. The surface of TFN membranes becomes smoother upon the addition of TNT in the PA layer, as demonstrated in Figure 4C–E. This is probably because of the long tubular structure of TNT that lays horizontally within the membrane PA matrix or on the membrane. This observation can be reasonably explained by the establishment of the intercalation between TNT and the PA network, which altered the morphology of the PA layer. A similar observation has been reported by Lai et al., who observed a flattened ridge and valley structure when introducing TNT nanofiller in the organic solution during interfacial polymerization [33]. The thickness of the PA layer was estimated from the FESEM images. For the TNT membranes, the thickness of the PA layer increased with the increasing TNT loading where the PA layer thickness of 0.01 TFN, 0.05 TFN and 0.1 TFN membranes was 0.318, 0.470 and 0.561 µm, respectively. The PA layer structure was covered by TNT layers that could be clearly seen within the PA layer as demonstrated in Figure 4C–E. The TNT was slightly intercalated with the PA network and resultant in the increase in the thickness of the PA layer. On the other hand, the finger-like structure of the PSf substrate was well preserved, suggesting that the overall PSf substrate was not altered by the modification at the PA layer. The sponge-like porous structure of the PSf sublayer is important to promote good water transport across the membrane. 

The presence of TNT in PA membrane was further confirmed by EDX mapping and spectra, as depicted in Figure 5. The observation for the peaks of carbon (C) (0.2 keV), oxygen (O) (2.3 keV), nitrogen (N) (0.3 keV) and sulfur (S) (2.3 keV) from the scattered signals provided the evidence of PA layer formation for all fabricated membranes. Meanwhile, the peaks of titanium (Ti) at 0.4 and 4.5 keV indicated the presence of TNT in the TFN membranes. The examination revealed the increment of Ti composition with 0.3%, 0.8% and 2.5% for 0.01 TFN, 0.05 TFN and 0.1 TFN membranes, respectively. 

Figure 6 demonstrates the three-dimensional AFM images and the corresponding roughness value for the TFC and TFN membranes. The topological characteristics of the membranes were consistent with the morphological observations from FESEM Images as presented in Figure 4. From the images, the bright and dark regions were related to the peaks and valley structures of the membrane surface. For TFC membrane, the mean roughness (Ra) and the peaks and valleys roughness (Rp-v) value of the PA membrane were 108.1 and 133.2 nm, respectively. Upon the incorporation of 0.01% TNT in the PA layer, the Ra and Rp-v values for 0.01 TFN membrane slightly declined to 78.17 and 98.8 nm, respectively. The Ra value of 0.05 TFN and 0.1 TFN membranes further decreased to 72.35 and 49.23 nm, respectively; meanwhile, the Rp-v value reduced to 91.32 and 63.25 nm, respectively. The observation evidenced that the introduction of TNT with different concentrations during interfacial polymerization considerably changed the membrane surface roughness. This observation implies that the interfacial crosslinking reaction on the growth of the leaf-like structure was successfully interfered by the embedment of TNT. A similar observation has been made by Yin et al. who reported the decrease in membrane surface roughness when the PA layer was embedded with graphene oxide [36]. These findings are in agreement with the surface images shown in Figure 6C–E, which revealed a rather smooth surface with the structure of TNT that lays horizontally in the PA layer (as can be seen in Figure 4C–E by FESEM analysis). Specifically, TNT formed hydrogen bond with TMC during IP at the reaction interface [37]. Consequently, the IP reaction rate was higher and the diffusion barrier was formed. When the growth of the PA layer was terminated prematurely, the active layer tended to be thinner and smoother. Therefore, it was noted that the TFN membranes are characterized by a smoother surface as compared to the unmodified TFC membrane. It is generally agreed that a smooth membrane surface can limit the surface–foulant interaction, hence suppressing membrane fouling.

The membranes were further analyzed with ATR-FTIR and XPS to obtain the respective chemical bonding and elemental composition details. Figure 7 depicts the XPS spectra of the TFC and TFN membranes. From the analysis, the peaks at approximately 284, 397 and 531 eV were attributed to O1s, N1s and C1s, respectively, implying the formation of the PA layer. Upon the modifications with TNT incorporated in the PA layer, the photoelectron peak of the Ti atom appears at a binding energy, Eb = 456.2 eV for Ti 2p^3/2^ and 461.9 eV for Ti 2p^1/2^, as well as for the O atom at 531.0 eV. However, the peak is not detected for 0.01 TFN membrane. The absence could be due to its low concentration in the membrane. Furthermore, a very low amount of Ti could be detected by EDX, as shown in Figure 6. The inset of Figure 7 shows the XPS spectra of 0.05 TFN and 0.1 TFN membranes. These peaks indicated the successful incorporation of TNT. The concentration of element O was much higher for TFN membranes compared to the TFC membrane. It is worth noting that the O element detected on TFN membranes’ surface arose from the hydrophilic OH group of TNT.

Figure 8a elucidates the ATR-FTIR spectra of all the membranes prepared. The peaks observed arose from the functional groups in both the PSf substrate and the PA layer [18]. The functional groups of PSf substrate were detected at 1500, 1290, 1240 and 1150 cm^−1^, which can be ascribed to the CH_3_-C-CH_3_ stretching, asymmetric O=S=O stretching, asymmetric C-O-C stretching, and symmetric O=S=O stretching, respectively. The three peaks found around 1652, 1611, and 1540 cm^−1^ were rendered by the functional groups of PA layer formed on the PSf substrate, as can be found for all TFN membranes as well as home-fabricated TFC and commercial TFC membrane. These peaks are related to the C=O stretching of amide I band, the N-H deformation of aromatic amide ring breathing, and the N-H bending of amide II band, respectively. Moreover, compared to TFC membrane, there was a small but observable change of the OH and Ti-O-Ti peaks for TFN membrane, possibly due to the TNT used in fabricating the composite membrane. These peaks with a slight difference in intensity suggested the embedment of TNT in the PA membrane.

Water contact angle is an important parameter of RO membrane, as it reflects the surface hydrophilicity, which can be related to the flux and productivity of the membranes. Figure 8b displays the water contact angle of TFC and TFN membranes. The surface contact angle of pristine TFC membrane was 46.1 ± 1.63°, which falls within the range of the typical PA layer [38]. The water contact angle of commercial TFC membrane is 28.3°, which is lower than the typical TFC membrane, as caused by the wetting properties of the rough TFC membrane surface via the Wenzel and Cassie effect. The presence of larger ridges and valley at the surface of the commercial TFC membrane contributes to the enhancement of water permeation by providing a more effective contact area between water molecules and membrane surface. Meanwhile, the water contact angle of TFN membrane dropped from 43.7° to 27.1° with the increasing of TNT concentration. The gradually decreased water contact angle suggested the enhancement in membrane surface hydrophilicity with the embedment of TNT within the PA layer. The TFN membrane surface was enriched with a higher amount of surface hydrophilic metal-OH functional group compared to pristine TFC membrane, as supported by ATR-FTIR analysis discussed earlier. The hydroxyl group on the tube surface of the nanomaterial strongly interact with the water molecules to facilitate water transport.

### 3.3. Membrane Separation Performance

#### 3.3.1. Salt Removal Efficiency

Figure 9a shows the separation performance based on the permeability and rejections for salt rejection. The salt removal efficiency was evaluated to ensure the intactness of the RO TFC and TFN membranes prepared in this study. At this stage, the TFN membrane separation performances were benchmarked with home-fabricated TFC membrane. From the observation, the home-fabricated TFC membrane achieved water permeability with ~0.67 L**·**m^−2^**·**h^−1^**·**bar^−1^. Upon the incorporation of TNT in the PA layer, the 0.01 TFN membrane demonstrated water permeability of 1.0 L**·**m^−2^**·**h^−1^**·**bar^−1^. However, with the TNT concentration increased up to ≥0.05% in the PA layer, the water permeability decreased rapidly to 0.5 L**·**m^−2^**·**h^−1^**·**bar^−1^. and 0.48 L**·**m^−2^**·**h^−1^**·**bar^−1^. for 0.05 TFN and 0.1 TFN membranes, respectively. 

The incorporation of TNT in the PA layer of the TFC membranes resulted in the improved water permeability, on account of the synergistic factors illustrated in Figure 9b. Firstly, the improvement can be associated to the enhanced membrane hydrophilicity. The TFN membrane surface was exposed with the abundant amount of hydrophilic functional group due to the presence of TNT (as verified from ATR-FTIR characterization). The improved membrane surface hydrophilicity facilitated the sorption of water on the membrane and improved the water permeability. The oxide surface reacts immediately with water molecules and the hydration layer is formed. Thus, it induced the water uptake and initiated a faster flow of water molecules to pass through the membrane. Secondly, the enhanced water permeability is also related to the additional water channel created by TNT. Nanotubular channels provide selectivity in molecular transport, making them a viable alternative to the typical solution-diffusion polymeric membrane. Thus, the unique water transport inside nanochannel of TNT evinced a bulk-like property water flow. Besides, the voids that were created from the interfacial gap between TNT and the PA matrix can also assist in transporting water. However, an excessive TNT incorporated in the PA layer remarkably reduced the water permeability. The consistent decrease in water permeability was due to the formation of additional mass transfer resistance of RO membrane. It can be deduced that the thickness of PA layer has a more dominant effect on the water permeability [39,40]. 

For the salt rejection performance, all TFC and TFN membranes showed a high rejection of >95%. The home-fabricated TFC membrane increased up to 95.92% for salt rejection. Upon the incorporation of TNT in the PA layer, 0.01 TFN membrane slightly dropped with 0.76% for the salt rejection. However, as the TNT concentration increased up to 0.05% and 0.1%, the salt rejection was increased to 96.74% and 96.1% for 0.05 TFN and 0.1 TFN, respectively. These observations clearly demonstrated the significant roles of the -OH and -COOH carboxyl groups on the membrane surface. The negatively charged TNT attracts sodium ions and the ionic adsorption forces that lead to the slight decline in salt rejection [41]. Besides, the increasing TNT concentration also tends to result in thicker TFN membrane. Specifically, the incorporation of TNT has sealed the defect pores of PA layer and led to the high rejection of salt.

#### 3.3.2. RO BPA and Caffeine Separation

Figure 10 presents the BPA and caffeine removal performance of TFN and TFC membrane. Based on Figure 10a, the commercial TFC membrane attained the lowest permeability of BPA and caffeine with 0.52 and 0.51 L**·**m^−2^**·**h^−1^**·**bar^−1^, respectively. Compared to commercial membrane, the home-fabricated TFC membrane exhibited improved permeability by 35.8% and 41.4%, respectively, for BPA and caffeine. The highest water permeability of 1.04 and 1.02 L**·**m^−2^**·**h^−1^**·**bar^−1^ for BPA and caffeine, respectively, has been achieved by 0.01 TFN membrane. However, downward trend of water permeability during the filtration of BPA and caffeine has been observed for 0.05 TFN and 0.1 TFN. 

The EDC separation performance of the membranes is governed by the characteristics of both membrane and EDC molecules [40,42]. Interestingly, the higher water permeability exhibited by home-fabricated TFC membrane as compared to commercial TFC membrane was mainly due to the thinner PA layer, as supported by the FESEM analysis. Besides, the reasons for the higher permeability of 0.01 TFN membrane during BPA and caffeine separation are similar to that observed for NaCl removal. The improved membrane hydrophilicity and interface void formation that resulted from the embedment of TNT in the PA layer has a significant impact on the high permeability of membrane performance. The poor adhesion between nanoparticle and polymer chains has caused the void formation around the nanoparticle [43]. However, an excessive use of TNT rendered negative effect towards the integrity of the TFN membrane. The increased PA layer thickness could be the main reason for the declined permeabilities for 0.05 TFN and 0.1 TFN membranes, as observed for NaCl removal.

The membranes tested in this study demonstrated a BPA rejection of >75% and a caffeine rejection of >93%. The commercial TFC membrane achieved BPA and caffeine rejections of 76.2% and 93.44%, respectively. Compared to the commercial TFC membrane, the home-fabricated TFC membrane demonstrated a higher rejection of BPA and caffeine of 89.3% and 95.87%, respectively. After the modifications have been made with TNT, the rejection performance was in the range between 89% and 92% for these 0.01 TFN, 0.05 TFN and 0.1 TFN membranes. Meanwhile, the rejection of the caffeine was increased to 97% and dropped to 95%. The 0.01 TFN membrane showed the higher rejection of caffeine. However, with the high concentration of TNT, 0.05 TFN and 0.1 TFN membrane showed a caffeine rejection of 96.06% and 95.5%, respectively. 

The BPA and caffeine rejection ability exhibited by the membranes is rendered by the steric effect induced by the membrane surface. The higher caffeine rejection compared to that of BPA is mainly due to the difference in their molecular sizes. There are various RO membranes with different micropollutants that showed that the size exclusion dominated the retention [7]. In fact, as the solute molecular weight higher than 100, the rejection efficiency will be higher than 90% [11]. The log of the octanol-water (logKow) of the EDC compounds (Table 1) is also known as a governing factor for the rejection trend [44,45]. The compounds with a logKow partitioning coefficient of less than 2.7 and 3.0 are considered as compounds with low hydrophilicity and high hydrophobicity, respectively [46]. The higher the logKow, the higher the rejection due to the hydrophobicity property of the solute towards the hydrophilicity property of the membrane [47]. Thus, the membranes incorporated with hydrophilic TNT can repel BPA from passing through the membrane. Other than that, a notable increment of BPA rejection could be attained by the adsorption of the BPA on the membrane via hydrogen bond interaction between the BPA molecule and membrane surface. Specifically, the OH group that attached to the aromatic end of BPA structure is bound to the high electromagnetic carbonyl functional group [48]. Thus, it can be concluded that the membrane could maintain its retention of BPA due to the higher number of possible sites of the hydrogen bonding formation. 

It was observed that caffeine, which is a hydrophilic organic compound, attained higher rejection than the hydrophobic BPA rejection for all membranes. In aqueous solution, caffeine molecules are solvated and the effective diameter is increased [49]. Therefore, the rejection of caffeine became more efficient compared to BPA rejection. However, as the TNT concentration was higher than 0.01% in the PA layer, the molecules can eventually diffuse inside the PA membrane towards the permeate side. It is due to the formation of detects on the PA layer of membrane with the interferences of TNT during interfacial polymerization. Furthermore, the hydrophilic caffeine can be more preferably absorbed on the membrane surface. Consequently, the rejection tends to decline, as happened for caffeine separation. In conclusion, among the membranes prepared, 0.01 TFN membrane has been identified as the most appealing membrane for BPA and caffeine removal.

#### 3.3.3. Antifouling Assessment on RO Membranes

It is essential to understand the fouling behaviors and the associated fouling mechanism during the filtration of BPA and caffeine to extend the usage of the membranes for practical applications. A long-term test with a duration of up to 540 min has been conducted for both TFC and 0.01 TFN membrane. As shown in Figure 11a, both TFC and 0.01 TFN membrane exhibited a significant permeability drop at the initial stage of BPA filtration, followed by an increase, and then remained constant. On the other hand, during the filtration of caffeine solution, as presented in Figure 11b, a similar trend was observed for TFC membrane, but, for 0.01 TFN membrane, the permeability dropped at the middle stage and became steady until the end of the experiment. However, 0.01 TFN membrane can sustain high permeability compared to TFC membrane throughout the experiment for both BPA and caffeine. The permeability drop of 0.01 TFN membrane for caffeine was possibly due to the physical adsorption and deposition of foulants on membrane surface. 

This fouling extent of RO membrane can be related to the hydrophilicity and roughness of the membrane surface. The enhanced hydrophilicity of 0.01 TFN membrane effectively weakened the hydrophobic attraction between organic foulants and membrane surface. For BPA, interaction has been established between the hydrophobic BPA and hydrophilic membrane surface. Meanwhile for caffeine, its hydrophilicity property was attracted to the hydrophilic membrane surface. Thus, 0.01 TFN membrane exhibited a slight drop in the caffeine permeability compared to TFC membrane but still maintained its higher permeability throughout the filtration duration. On the other hand, the antifouling performance trends were based on the relationship between the membrane surface roughness and antifouling performance. Firstly, it was noticed that the hydrophilic surface of 0.01 TFN membrane makes it easier to establish a water layer on the membrane surface. Besides, the significant smoothening of the membrane surface also might reduce the tendency of the solute attachment, as the ridge and valley structure of the TFC surface has greater affinity towards hydrophobic foulants through mechanical interlocking.

#### 3.3.4. Benchmarking with Other Studies

Table 2 summarizes the results from the past and current investigations. Overall, RO is a promising process to remove EDC. There are studies that utilized commercial RO membranes to remove BPA [4,11]. The result that obtained was a fall in the range of 74–87%. Licona et al. used the BW30 of RO membrane to eliminate caffeine compound and reported a removal of 97% [7]. Mat Anan et al. incorporated TiO_2_ nanoparticle within TFC membrane and reported an excellent BPA rejection of 99% with 100mg/L feed concentration [40]. They concluded that the rejection was governed by the physicochemical properties of solute and the membrane surface. Thus, the understanding of the interaction is essential for the improvement and selection of suitable membrane. Comparing the present study with the work reported by Mat Anan et al., the TNT incorporated TFN membranes could achieve 89% rejection of BPA with a very-low-concentration BPA of 10 mg/L. The findings reveal that the nanotubular channel in the PA layer of RO membrane could play a significant role in enhancing the water permeability while maintaining BPA rejection ability. The recyclability of the membranes is an important aspect in evaluating the practicability of a newly developed real wastewater treatment application. Hence, the recyclability of PA/TNT TFC will be evaluated in our future study.

## 4. Conclusions

This study explored the construction of water channels within the PA layer of TFC RO membrane for effective BPA and caffeine removal. The best membrane prepared in this study, 0.01 TFN, achieved 89.23% and 97% for BPA and caffeine removal with a permeability of >1 L**·**m^−2^**·**h^−1^**·**bar^−1^ for both BPA and caffeine filtration. The incorporated TNT enhanced the surface wettability by providing nanochannels extending through the PA layer. The orientations of TNT in the PA layer have smoothened the membrane surface, and the interface gaps between the TNT and PA layer allowed for water passage. These properties appear to be the most prevalent mechanism in controlling the permeability and rejection of BPA and caffeine. The long-term assessment indicated that the improved hydrophilicity and smooth surface rendered membrane antifouling properties. It is anticipated that the TFN membranes proposed in this study are also applicable for the removal of various types of EDC and pharmaceutically active compounds.

## Figures and Tables

**Figure 1 membranes-12-00958-f001:**
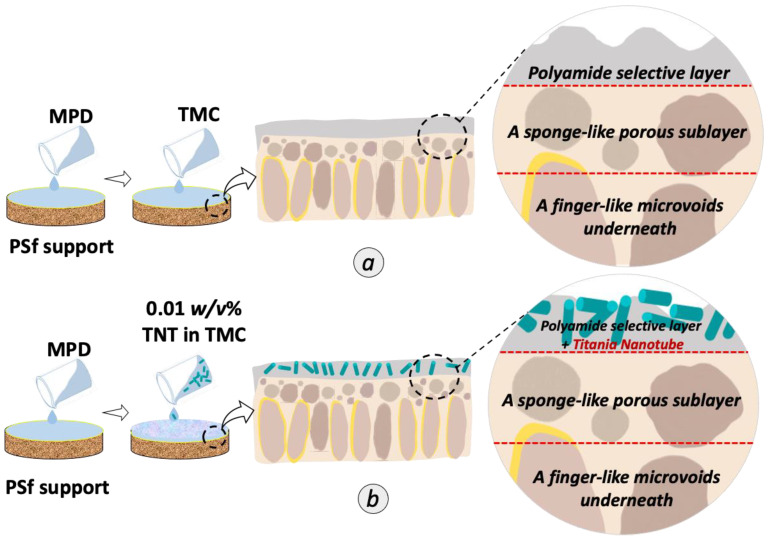
Schematic illustration of the preparation of (**a**) TFC and (**b**) TFN membranes incorporated with randomly aligned TNT in the PA selective layer.

**Figure 2 membranes-12-00958-f002:**
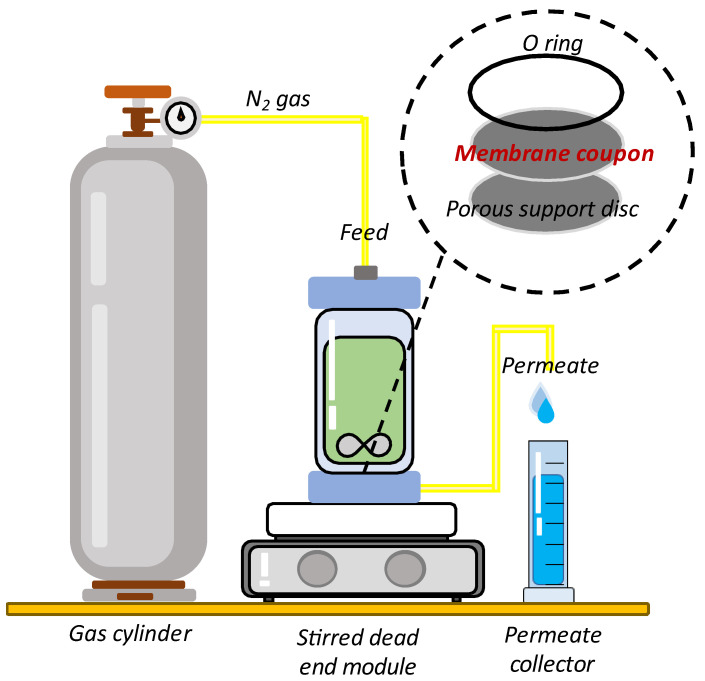
Schematic diagram of dead-end filtration system; the feed flow is forced through the membrane at a given pressure through nitrogen gas.

**Figure 3 membranes-12-00958-f003:**
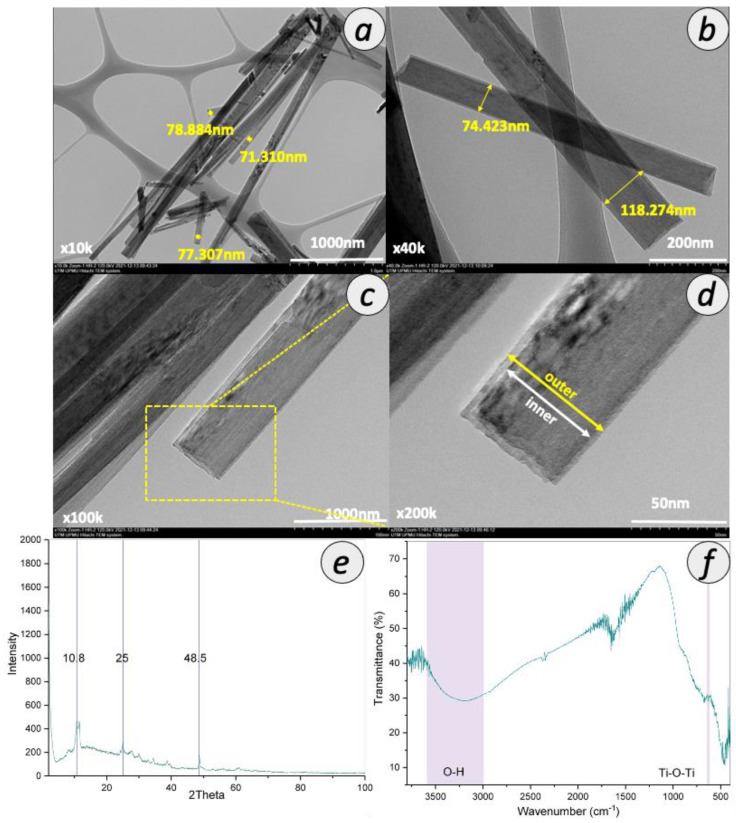
Properties of synthesized TNT with TEM image with magnification of (**a**) 10,000×, (**b**) 40,000×, (**c**) 100,000×, (**d**) 200,000×, (**e**) XRD spectra and (**f**) ATR-FTIR spectra.

**Figure 4 membranes-12-00958-f004:**
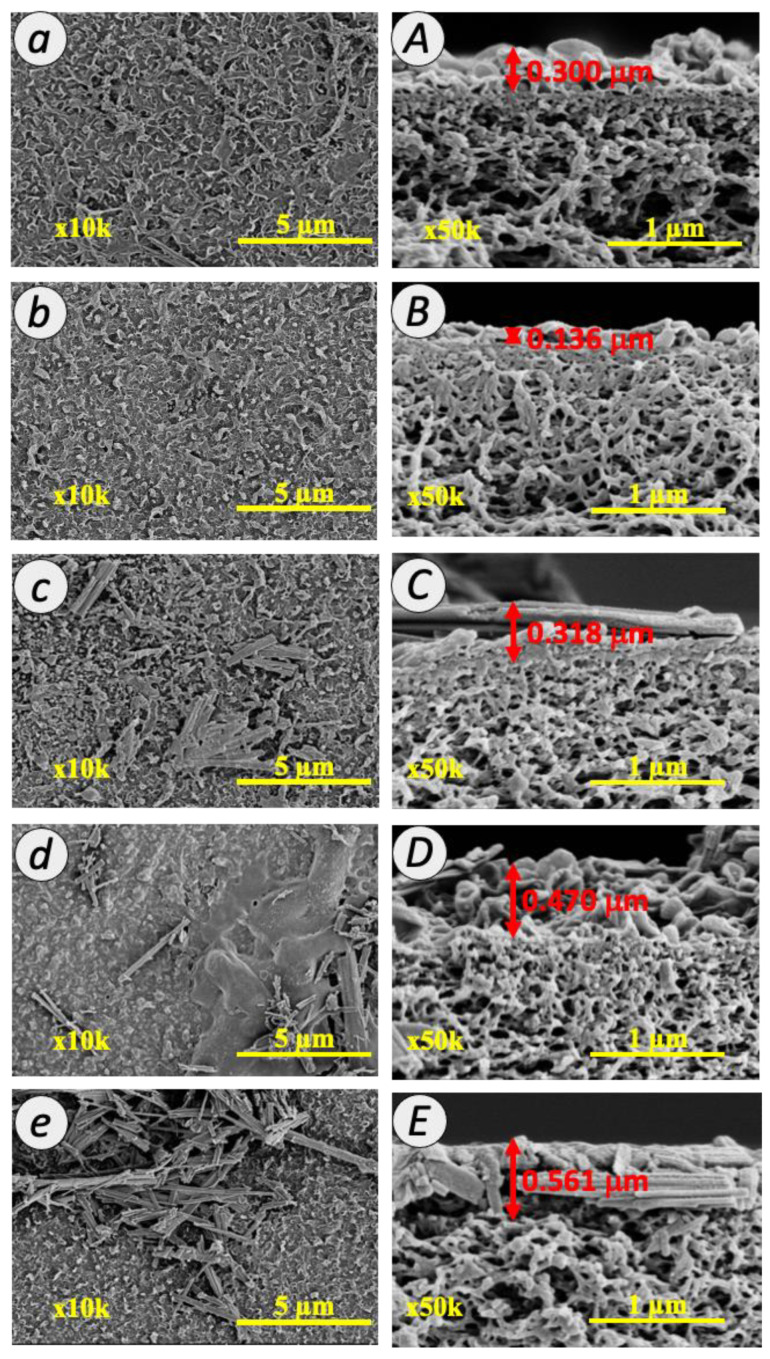
FESEM images for surface (**a**–**e**) and cross-section (**A**–**E**) of commercial TFC, TFC, 0.01 TFN, 0.05 TFN and 0.1 TFN membranes at 10 k and 50 k magnifications.

**Figure 5 membranes-12-00958-f005:**
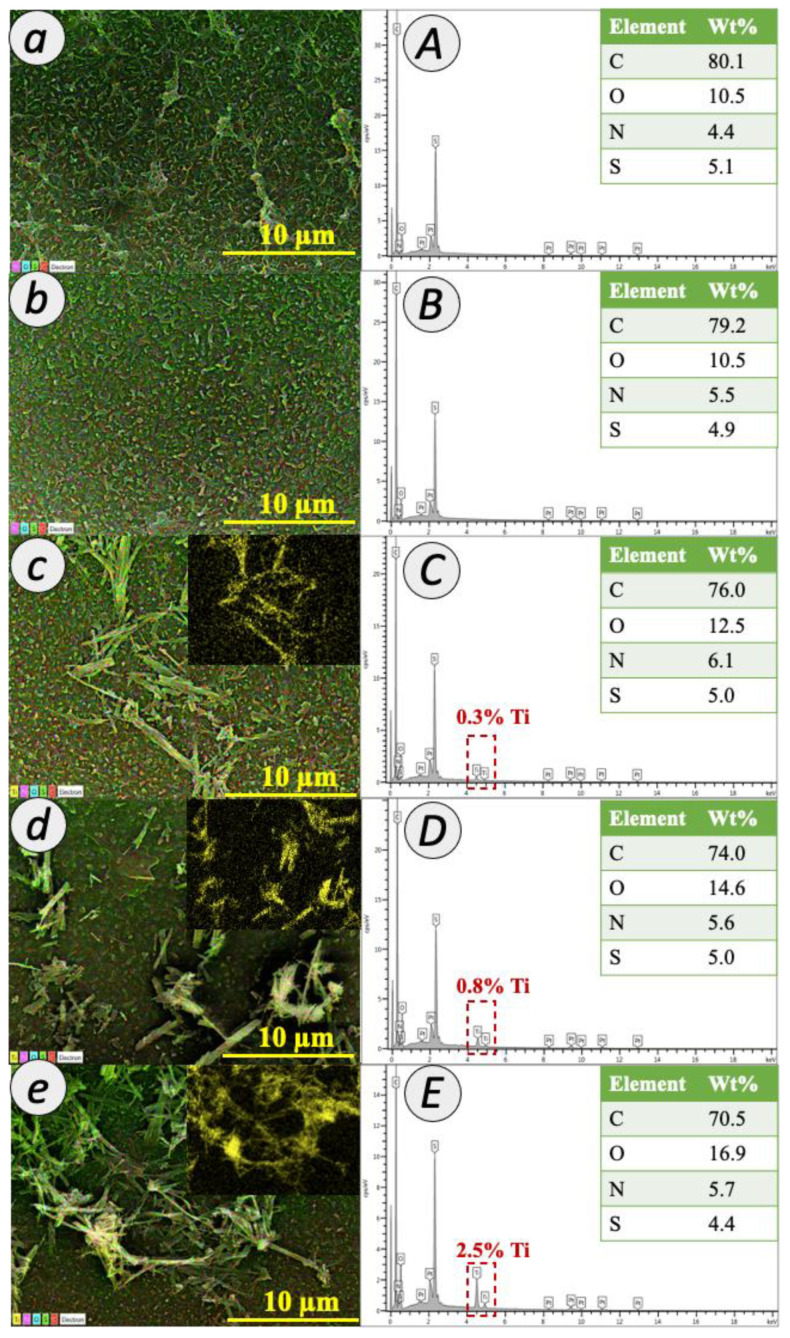
EDX mapping (**a**–**e**) and composition (**A**–**E**) of TFC and TFN membranes.

**Figure 6 membranes-12-00958-f006:**
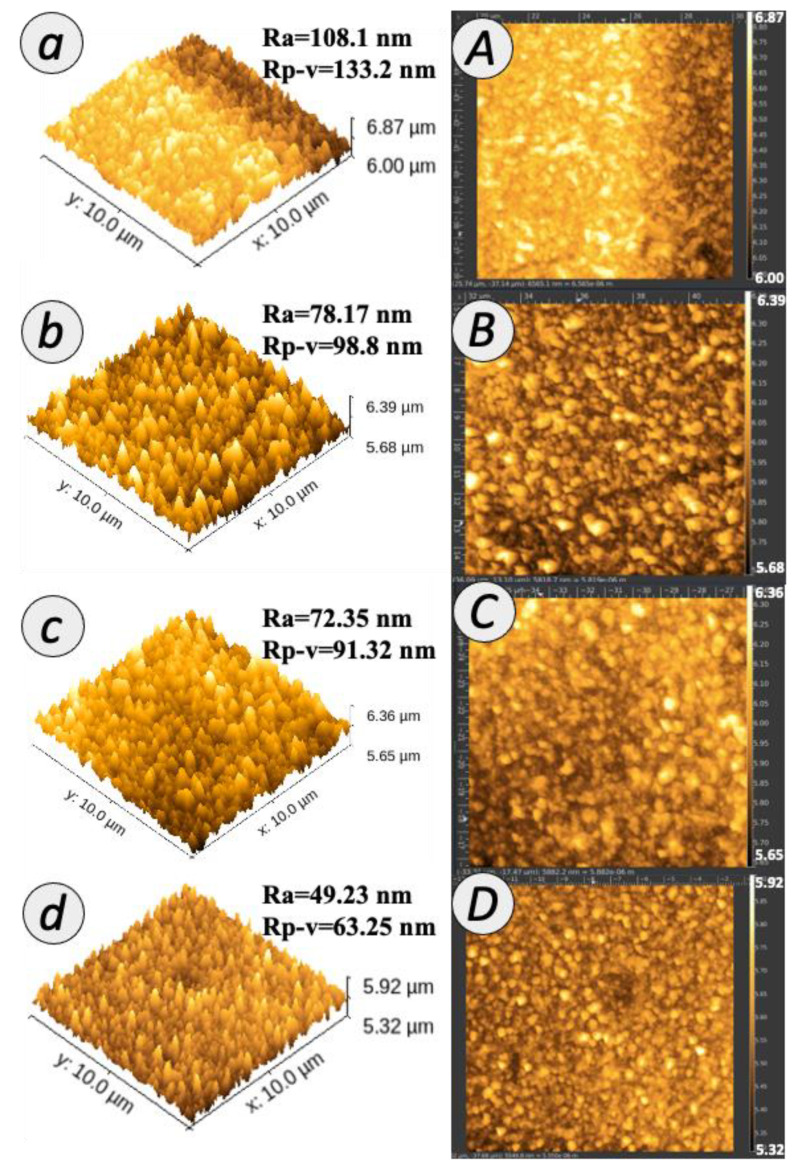
AFM of membrane surface with (**a**–**d**) three- and (**A**–**D**) two-dimensional images.

**Figure 7 membranes-12-00958-f007:**
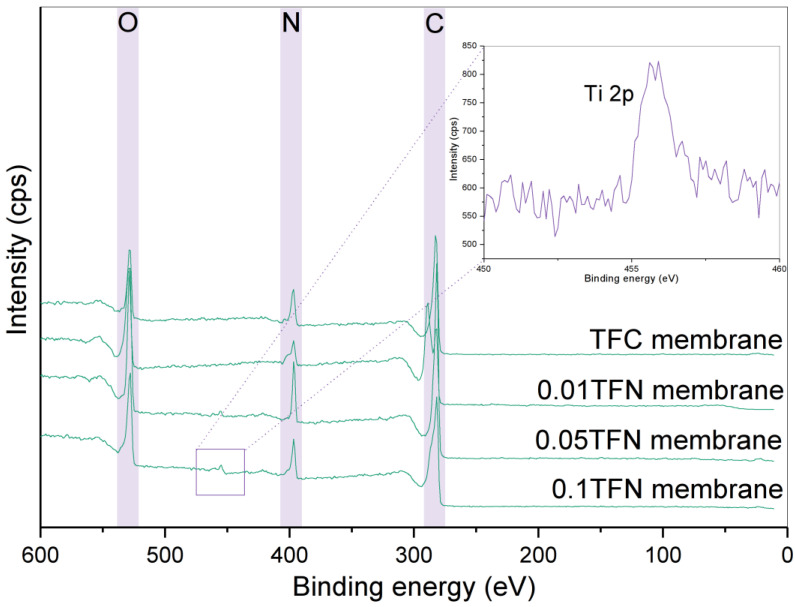
XPS spectra of TFC and TFN membranes (inset: the magnified XPS spectrum of the Ti 2p region).

**Figure 8 membranes-12-00958-f008:**
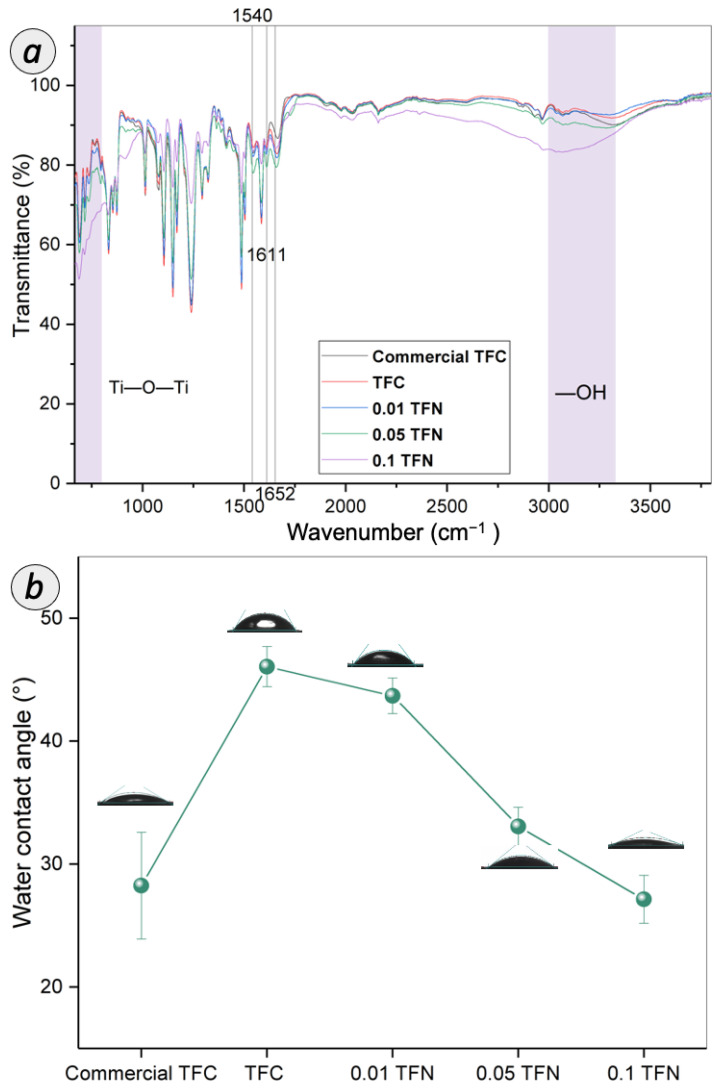
(**a**) ATR-FTIR spectra and (**b**) water contact angle of TFC and TFN membranes.

**Figure 9 membranes-12-00958-f009:**
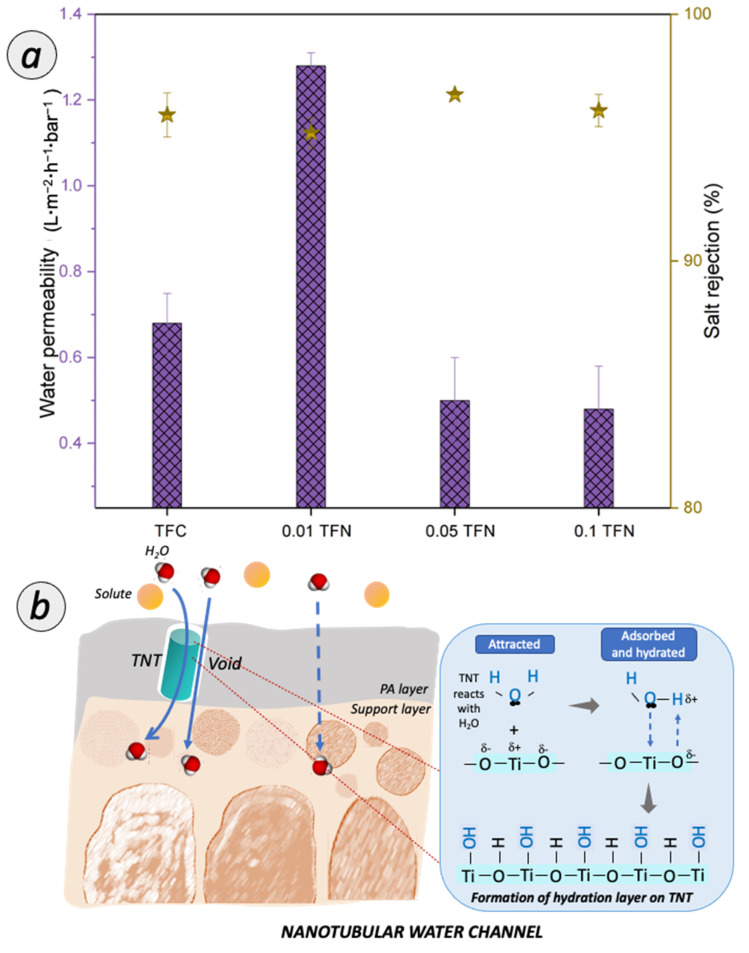
(**a**) Permeability and rejection of TFC and TFN membranes for desalination; (**b**) Possible pathway of water transport for TFN membrane.

**Figure 10 membranes-12-00958-f010:**
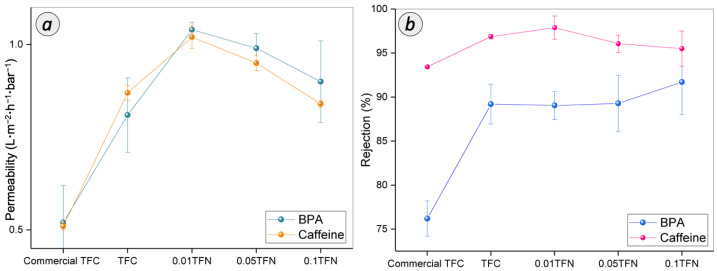
(**a**) Permeability and (**b**) rejection of TFC and TFN membranes for BPA and caffeine separation.

**Figure 11 membranes-12-00958-f011:**
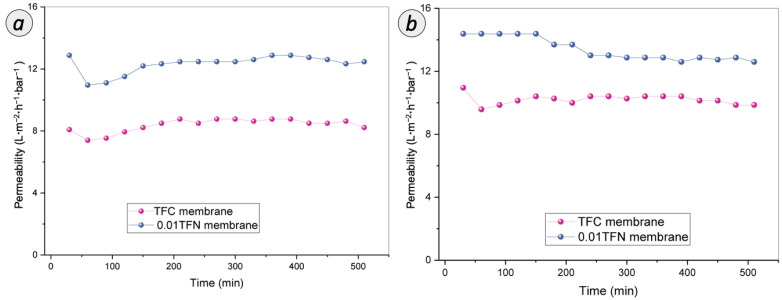
Long-term performances of TFC and 0.01 TFN membranes for the filtration of (**a**) BPA and (**b**) caffeine over duration of 540 min.

**Table 1 membranes-12-00958-t001:** Characteristics of model organic micropollutant used as EDC [7,27,28].

Name	Structure	Therapeutic Class	Molecular Weight g/mol	LogKow
BPA	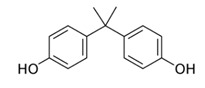	Cytotoxic therapy	228.0	3.32
Caffeine	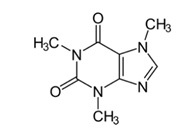	Central Nervous System stimulant	194.2	0.16

**Table 2 membranes-12-00958-t002:** Comparison of pristine TFC and modified TFN membrane performances based on their rejections.

Membrane	Feed Concentration (mg/L)		Rejection (%)		Ref.
	BPA	Caffeine	BPA	Caffeine	
Commercial RO (TW30–1812−100)	50	-	87	-	[4]
Commercial RO (CE BWRO)	1000	-	74–84	-	[11]
Commercial RO (BW30)	-	10	-	92–95	[7]
PA/TiO_2_ TFC	100	-	99	-	[40]
PA/TNT TFC	10	10	89.05	97.89	This study

## Data Availability

The data presented in this study are available on request from the corresponding author.

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
