# Peer review of "Enhanced Removal of Endocrine-Disrupting Compounds from Wastewater Using Reverse Osmosis Membrane with Titania Nanotube-Constructed Nanochannels"

_membranes, 2022, doi:10.3390/membranes12100958_

Round 1

Reviewer 1 Report

Dear authors,

Please find the enclosed file.

Author Response

Comments to Authors:

Reviewer 1

Dear authors, The manuscript provides a comprehensive data about RO membranes, please find the comments below to improve the article.

Thank you for the comments. All of the changes made are marked in the main text with red colour.

Minor correction

Comment 1

Line 1: Author should clearly mention the type of the manuscript. Authors should carefully check the names of authors and all the grammatical errors and typo mistakes should be removed.

The name of authors has been carefully checked as well as the grammatical errors and typo mistakes

Comment 2. Introduction:

  • Line 67: (below several µg/L) authors should provide some numerical value here.

Numerical value has been stated in line 68-69.

  • Line 87: Khoo et al authors should provide reference number here.

Reference number has been stated in the main text in line 94

  • Authors should provide the novelty of the work keeping in view the initial lines of abstract.

The novelty of this study has been highlighted in the initial line of abstract.

  • Authors should also provide a brief account of nanotubes other than titania which have been used in the literature and future potential target materials as well.

A brief account of nanotubes other than titania and future potential target materials have been provided. The description can be found in line 78-84.

Comment 3: Methodology

  • Line 155 authors should provide reference to calculate O/N ratio?

The references have been provided in line 169.

  • Line 173: 2,000 ppm of salt solution, authors should mention chemical nature of salt here. (c) Table 1. Units of molecular weight are missing.

The chemical nature of salt, NaCl has been mentioned and unit of molecular weight has been stated in Table 1

  • Table 1: The digits after the decimals should be similar.

The digit after the decimals has been similarly stated.

  • Line 205: t is the time treatment, should be time or duration of treatment.

The time treatment has been changed to duration of treatment for t.

Comment 3: Results

  • This section should be results and discussion

The section name has been changed to result and discussion

  • Line 217-224: All synthetic information should be in experimental part. Whereas the general information such as “alkali solution (NaOH) is used as a ‘bridging agent’ to 220 attach TiO2 nanoparticles together“ should be included in introduction.

The line 217-224 has been moved to the methodology and can found in line 123-134.

  • Figure 1: It needs revision for better understanding. The caption “a” part should be elaborated. In figure MPD and TMC looks same hard to extract what is happening at this stage. The font size is small throughout the figure.

The elaborations have been made in the description part (line 138-148). The font size in Figure 1 has been increased.

  • Figure 3c, X-axis should be 3500-500 cm-1 .

The x-axis in Figure 3c has been changed to 3500-500cm-1

  • Line 240-243, authors should reconsider the statement and provide justification of variation in area for their product.

The statement has been justified with some literatures and can be found between the line 249-258.

  • Line 248-249: The inner diameter of TNT was determined as 19.53 nm, in agreement with that of reported previously.

The inner diameter from literatures have been stated in line 256-257.

  • The captions of Figures 2 and 6 should be elaborated.

Captions of Figures 2 and 6 have been elaborated

Reviewer 2 Report

    A reverse osmosis membrane with titania nanotube-constructed nanochannels was prepared and used for the removal of endocrine disrupting compounds. The paper is well designed and organized. Satisfactory results have been obtained. I recommend a minor revision for publication.   1: In the abstract, EDC should be introduced as EDCs. Please check and revise. 2: The service life or using cycles should be introduced in the abstract and the text. 3: Please check Table 1. The first line and the last row should be aligned.  4: In Figure 4, too many photos crowded together. There should be white space between images. 5: In Figure 7, only one line was amplified, but there were two lines that were boxed. 6: In Figure 8, it is hardly to see clear the curves in fig (a).  7: In Figure 9 (a), please use a block diagram instead of a line diagram.  8: How many times could the proposed membrane be used? How about the RSD? It should be added in the performance evaluation.  9: Both in the title and in the conclusion, the author emphasized the enhanced performance. How did the membrane enhance the removal of EDCs? More explanation should be added. 10: In line 533-537, the unit of ppm could not be used.  11: Why only BPA and caffeine were used as model EDCs? How about the result if other analytes were tested?

1: In the abstract, EDC should be introduced as EDCs. Please check and revise.

2: The service life or using cycles should be introduced in the abstract and the text.

3: Please check Table 1. The first line and the last row should be aligned.

4: In Figure 4, too many photos crowded together. There should be white space between images.

5: In Figure 7, only one line was amplified, but there were two lines that were boxed.

6: In Figure 8, it is hardly to see clear the curves in fig (a).

7: In Figure 9 (a), please use a block diagram instead of a line diagram.

8: How many times could the proposed membrane be used? How about the RSD? It should be added in the performance evaluation.

9: Both in the title and in the conclusion, the author emphasized the enhanced performance. How did the membrane enhance the removal of EDCs? More explanation should be added.

10: In line 533-537, the unit of ppm could not be used.

11: Why only BPA and caffeine were used as model EDCs? How about the result if other analytes were tested?

Author Response

Comments to Authors:

Reviewer 2:

A reverse osmosis membrane with titania nanotube-constructed nanochannels was prepared and used for the removal of endocrine disrupting compounds. The paper is well designed and organized. Satisfactory results have been obtained. I recommend a minor revision for publication.

Thank you for the comments. All of the changes made are marked in the main text with purple colour.

  • In the abstract, EDC should be introduced as EDCs. Please check and revise.

EDCs have been introduced in the abstract.

  • The service life or using cycles should be introduced in the abstract and the text.

The recyclability of the membrane is not included in this scope of this manuscript. The main aim of this study is to evaluate the feasibility of our newly developed membrane in treating EDC at low concentration. The study related to the recyclability of the newly developed membrane is mentioned as recommendation for our future study in line 544 to 547.

3          Please check Table 1. The first line and the last row should be aligned.

The first line and the last row have been aligned

4          In Figure 4, too many photos crowded together. There should be white space between images.

Figure 4 has been improved

5          In Figure 7, only one line was amplified, but there were two lines that were boxed.

Figure 7 has been improved

6          In Figure 8, it is hardly to see clear the curves in fig (a).

Figure 8 has been enlarged and improved

7          In Figure 9 (a), please use a block diagram instead of a line diagram.

            Figure 9(a) has been improved with a block diagram.

8          How many times could the proposed membrane be used? How about the RSD? It should be added in the performance evaluation.

The recyclability of the membrane is not included in this scope of this manuscript. The main aim of this study is to evaluate the feasibility of our newly developed membrane in treating EDC at low concentration. However, based on the reviewer’s suggestion, the reusability of the optimized membrane will be evaluated on our future study

9          Both in the title and in the conclusion, the author emphasized the enhanced performance. How did the membrane enhance the removal of EDCs? More explanation should be added.

The removal of EDC has been explained in the line 455 to 495

10        In line 533-537, the unit of ppm could not be used.

            The unit of ppm has been changed to mg/L

11        Why only BPA and caffeine were used as model EDCs? How about the result if other analytes were tested?

Thank you for pointing this out. Currently there are many EDC that can be found in wastewater. However, this study focuses on BPA and caffeine due to their high concentrations in most of the rivers in Malaysia. The justification of selection has been provided in line 98-99

Reviewer 3 Report

Waste water treatment is important for water safty. The topic of manuscript is of broad interesting to readers. The experiments are well designed and the manuscript is well organized. Minor revision is suggested by solving the following issues.

1.     Various materials are developed for waste water treatment. Some typical references are recommended: A review on conversion of crayfish-shell derivatives to functional materials and their environmental applications; Synthesis and Application of Granular Activated Carbon from Biomass Waste Materials for Water Treatment: A Review; MOFs meet wood: Reusable magnetic hydrophilic composites toward efficient water treatment with super-high dye adsorption capacity at high dye concentration.

2.     Please pay attention to the writing of subscripts and superscripts.

3.     The units should be written in the same style. “mg/L”, “m2/g” and “L.m-2.h-1.bar-1” are different styles.

4.     In line 239, “The surface area of TNT was analysed as 33.58 m2/g.” N2 absorption/desorption isothermal should be offered.

5.     It is hard to tell the tubular structure of TNT from SEM and TEM images. The authors should give other evidences to confirm it.

6.     Please double check the references. Page numbers are missing for some references.

Author Response

Comments to Authors:

Reviewer 3:

Waste water treatment is important for water safety. The topic of manuscript is of broad interesting to readers. The experiments are well designed and the manuscript is well organized. Minor revision is suggested by solving the following issues.

Thank you for the comments. All of the changes made are marked in the main text with green colour.

  • Various materials are developed for waste water treatment. Some typical references are recommended: A review on conversion of crayfish-shell derivatives to functional materials and their environmental applications; Synthesis and Application of Granular Activated Carbon from Biomass Waste Materials for Water Treatment: A Review; MOFs meet wood: Reusable magnetic hydrophilic composites toward efficient water treatment with super-high dye adsorption capacity at high dye concentration.

The references have been referred and cited in line 74 .

  • Please pay attention to the writing of subscripts and superscripts.

The subscripts and superscripts have been carefully stated.

  • The units should be written in the same style. “mg/L”, “m2/g” and “L.m-2.h-1.bar-1” are different styles.

The different units that stated were depends on their measurement types.

  • In line 239, “The surface area of TNT was analysed as 33.58 m2/g.” N2 absorption/desorption isothermal should be offered.

N2 absorption/desorption isothermal has been provided in Supporting information as Figure S1.

  • It is hard to tell the tubular structure of TNT from SEM and TEM images. The authors should give other evidences to confirm it.

The tubular structure can be confirmed as shown in Figure 3a(iii-iv) and stated in line 234 235

  • Please double check the references. Page numbers are missing for some references.

The references have been checked.
